# Traditional and Undulating Periodization on Body Composition, Strength Levels and Physical Fitness in Older Adults

**DOI:** 10.3390/ijerph19084522

**Published:** 2022-04-08

**Authors:** Salvador Vargas-Molina, Manuel García-Sillero, Ramón Romance, Jorge L. Petro, José Daniel Jiménez-García, Diego A. Bonilla, Richard B. Kreider, Javier Benítez-Porres

**Affiliations:** 1Faculty of Sport Sciences, EADE-University of Wales Trinity Saint David, 29018 Málaga, Spain; salvadorvargasmolina@gmail.com; 2Physical Education and Sport, Faculty of Medicine, University of Málaga, 29016 Málaga, Spain; benitez@uma.es; 3Laboratory Fivestars, 29018 Málaga, Spain; 4Body Composition and Biodynamic Laboratory, Faculty of Education Sciences, University of Málaga, 29071 Málaga, Spain; arromance@uma.es; 5Research Division, Dynamical Business & Science Society—DBSS International SAS, Bogotá 110311, Colombia; jlpetro@dbss.pro (J.L.P.); dabonilla@dbss.pro (D.A.B.); 6Research Group in Physical Activity, Sports and Health Sciences (GICAFS), Universidad de Córdoba, Montería 230002, Colombia; 7Department of Health Sciences, Faculty of Health Sciences, University of Jaén, 23071 Jaén, Spain; josedanieljimenezgarcia@gmail.com; 8Sport Genomics Research Group, Department of Genetics, Physical Anthropology and Animal Physiology, Faculty of Science and Technology, University of the Basque Country (UPV/EHU), 48940 Leioa, Spain; 9Exercise & Sport Nutrition Lab, Human Clinical Research Facility, Texas A&M University, College Station, TX 77843, USA; rbkreider@tamu.edu

**Keywords:** aging, resistance training, aged, elderly, muscular function, functional capacity

## Abstract

Introduction: Undulating training has been investigated in sedentary and trained adults, but less is known about the influence of undulating training in older adults. Purpose: This study aimed to evaluate body composition, strength levels, and physical fitness in response to traditional or undulating training in older adults. Methods: A controlled, double-arm trial was conducted in eighteen older adults (10 males, 8 females; 64 ± 2.1 years; 165.12 ± 7.5 cm; 72.5 ± 11.4 kg; 26.5 ± 3.2 k·gm^−2^) who were randomly assigned to traditional (*n* = 9, TT) or undulating training (*n* = 9, UT) for eight weeks. Dual X-ray absorptiometry was used to measure fat-free mass (FFM), fat mass (FM), and bone mineral density (BMD). Strength levels were evaluated by the handgrip strength and the one-repetition maximum in vertical chest press, rowing machine, squat, monopodal horizontal leg press, and leg extension. In addition, functional capacity was assessed using the Senior Fitness Test (SFT). Statistical analysis included mean/median comparisons to establish the difference after the intervention (paired Student’s *t*-test or Wilcoxon test), and effect size calculations based on estimates. Results: After correction for fat-free adipose tissue, a significant increase in FFM was observed in both groups, while no significant changes were found in FM and BMD. Upper- and lower-limbs strength showed significant increases in both groups, although clinical significance varied among exercises. Favorable results were seen on the cardiorespiratory fitness and strength components of the SFT in both groups. Conclusions: The 8-week UT and TT protocols are valid options for improving FFM and increasing strength and functional capacity in women and men over 60 years of age.

## 1. Introduction

The aging process leads to a progressive reduction in strength [1] and fat-free mass (FFM) [2]. Importantly, it is associated with an increase in fat mass (FM) and reduction in bone mass, which is often referred to as sarcopenic obesity [3,4]. This physical deterioration may negatively affect the functional dependence of healthy old adults (HOA) [5], which may increase the likelihood of falls [6]. It has been shown that strength training induces positive adaptations in physical functionality, bone mineral density, and metabolic control [7]. For example, improvements in balance, FFM, and strength levels [8] cause a positive effect on fall prevention [9,10,11]. It is noteworthy that the muscle action at maximal-intended velocity against a force, such as resistance training (RT), might increase upper- and lower-limbs muscle power better than traditional RT [12]. On the other hand, the loss of muscle power is associated with an increased risk of falls and a decrease in functional capacity, interfering negatively with quality of life [13,14]. 

It is clear that physical strength can be improved in several ways, but current evidence suggests that rate of force development (RFD) should be emphasized as it has been proven to be the most optimal method for improving activities of daily living and the reduction of falls in HOA [15,16]. In this regard, manipulation of programming variables can optimize results, such as a frequency of 2–3 training days per week [16], the application of maximal intended velocity [17,18], and a volume of one to three sets per exercise [19]. In terms of work intensity, conflicting results have been found; for example, higher increases in strength levels are generated at 70–85% of the repetition maximum (RM), whereas changes in muscle morphology and functional performance are optimized around 50–70% RM [16]. For this reason, more research is needed to optimize the organization of RT work in older adults. The largest body of research has focused on organizing programming variables in traditional training (TT), where the same range of repetitions and intensity is executed throughout the entire research phase. Notwithstanding, undulating training (UT), where different loads and repetition ranges are applied daily or weekly [20], has been less studied. A recent systematic review found no improvements in strength levels in favor of any protocol, linear or non-linear [21]. In fact, the unique study that has evaluated the TT versus UT program in older participants revealed no advantage in favor of either protocol on neuromuscular or functional parameters [22]. Since UT might be an alternative for increasing and/or maintaining body composition, strength levels, and functional capacity, the aim of this study was to compare the effect of eight weeks of UT and TT training on strength, body composition, and physical fitness parameters in HOA. We hypothesized that UT training for eight weeks results in equal or superior improvements in body composition, strength, and functionality than traditional training in older adults.

## 2. Methods

### 2.1. Trial Design

This study was conducted as a double-arm and repeated-measures randomized clinical trial in older adults. The initial sample was 24 participants, and, finally, 18 participated and were randomized in a 1:1 fashion to either the UT (*n* = 9) or TT (*n* = 9) protocol. After three weeks of familiarization, participants performed three exercise sessions per week with 48 h of recovery for eight weeks. The UT group performed different ranges of repetitions in each session while the TT protocol had a fixed range in all sessions.

### 2.2. Participants

Twenty-four older adults with no previous experience in overload training were potentially eligible to participate in this study. Participants were informed about the possible risks of the experiment and signed an informed consent form. The research protocol was approved by the Ethics Committee of the University of Malaga (code: 38-2019-H) in accordance with the ethical guidelines of the Declaration of Helsinki [23]. Participants who reported cardiac conditions, hyperglycemia, hypertension, and/or osteo-articular problems (mainly in the hip and/or knee) were excluded, as well as those who reported any type of problem that prevented the completion of the program. Individuals who were currently immersed in a physical exercise program prior to three months were also excluded. The required age to participate was established between 58 and 65 years old. Additionally, a medical specialist’s report was required indicating the lack of pathologies that would prevent incorporation into the exercise program.

### 2.3. Dietary Intake

All participants were instructed to consume a high-protein diet (2 g·kg^−1^ FFM·day^−1^) both in the familiarization phase and during the investigation. A nutritionist guided food choices, but no strict record of dietary intake was carried out.

### 2.4. Anthropometry

All anthropometric data were collected during the first visit to the laboratory during the familiarization period. Body mass was measured with a digital scale to the nearest 50 g (Tanita RD-545, Tokyo, Japan). A fixed stadiometer was used to measure the stature (SECA 220, Hamburg, Germany).

### 2.5. Body Composition

Total body mass and regional body composition were estimated using dual X-ray absorptiometry (APEX 3.0 software version, Hologic QDR 4500, Bedford, MA, USA). For each scan, participants wore light clothing and were asked to remove all materials that could attenuate the X-ray beam, including jewelry items and underwear containing wire. The coefficient of variation was less than <1.5% for all measurements of segmental and whole-body body composition, including bone mineral density (g·cm^−2^), mineral content (g), FM (%), FM (g), FFM (g), and total body mass (g). The DXA was calibrated with phantoms according to the manufacturer’s guidelines each day before measurement. Assuming that 85% of adipose tissue is fat, the fat-free adipose tissue (FFAT) was estimated with the equation (FM/0.85) × 0.15 [24]. From this, the FFM-FFAT was calculated and reported as our group has done recently [25].

### 2.6. Functional Capacity

One-repetition maximum (1 RM) strength assessments were performed on 2 days with 72 h between sessions. Participants had 1 RM determined on all upper and lower limb exercises, as shown in Figure 1. Tests were alternated from upper to lower body to avoid fatigue and overloading of the muscle areas used. All the tests described were carried out on Gervasport machines (Gervasport, Madrid, Spain). For this purpose, the correct technique was explained to the participant by means of modeled and verbal instruction. The maximum prehensile handgrip strength (HGS) in both hands was measured with a hand-held dynamometer (GRIP-D TKK 5401, Takei Scientific Instruments CO, Tokyo, Japan). Hand dominance was determined by asking the participant. The protocol recommended by the American Society of Hand Therapists (ASHT) was performed [26].

The Senior Fitness Test (SFT) was also performed to assess functional capacity. This battery of tests includes: 1 RM-Arm Curl test (ACT), Back Scratch test (BST), Chair sit and reach test (CSRT), Chair stand test (CST), 6-min six walk test (6 MWT), and the 8 Foot Up-and-Go Test (8-FUG). All procedures were performed according to previous recommendations [27]. In addition, the 1RM-Row machine test (RM) was applied. These assessments were performed in the first week of familiarization in conjunction with the other tests.

### 2.7. Familiarization

All training and familiarization sessions were supervised by the research team. This phase lasted three weeks, with three sessions per week, and with 48 h of recovery between sessions. Participants were informed of the training protocol to be followed and the loads were adjusted for all exercises. Additionally, a nutrition specialist informed them of the dietary guidelines to be adopted. 

### 2.8. Exercise Protocol

The TT training protocol was performed with three sets of 8–10 RM, adjusting the loads so that the participants did not reach volitional failure, between 2–3 repetitions in reserve (RIR, “reps in reserve”), with pauses of 1.5 min between sets and exercises. Moreover, the UT protocol was performed for the first session of the week with a repetition range of 3–5 RM, with 3 min pause; the second day at 8–10 RM, with 1.5 min pause; and the third day at 20 RM, with 45 s, performing a daily undulation. Both protocols were performed with the maximum movement velocity, both the eccentric and concentric phases, as long as there were no deficiencies in the techniques, since it can promote greater functional improvements than resistance training performed at slower speeds in older adults [17,18]. In addition, both protocols used the same exercises, as shown in Figure 1.

In all the sessions, two researchers supervised and adjusted the loads of each exercise according to the instructions of the participants. For this, 3 weeks of familiarization were carried out, to record the loads, the sensations of effort, and the repetitions in reserve. Thus, it was self-adjusted in all sessions and for each participant.

### 2.9. Statistical Analysis

Results are expressed as mean, standard deviation (SD), and 95% confidence interval (95% CI). The normality of the data was evaluated with the Shapiro–Wilk test and the homogeneity of variances with Levene’s test. Comparison between the variables at baseline and change (Δ = post-test–pre-test) was performed with the Student’s t-test for independent samples or the Mann–Whitney U test, while the intra-group (pre-test vs. post-test) comparison was performed using the Student’s t-test for paired samples or the Wilcoxon test, and effect sizes (ES) were calculated with Hedges’ g. The above procedures were performed with the software SPSS version 25 (IBM Corp, Armonk, NY, USA), assuming a significance level of 0.05 for all tests. In addition, the effect size was computed with the R package Data Analysis using Bootstrap-Coupled Estimation (DABEST) v0.3.0 [28] within the R statistical computing environment version 4.0.0 [29].

## 3. Results

### 3.1. Demographics

A total of eighteen older adults (5 male and 4 female per protocol, 64 ± 2.1 years, 165.12 ± 7.5 cm, 72.5 ± 11.4 kg, 26.5 ± 3.2 k·gm^−2^) completed the intervention program and were included in the statistical analysis. Six subjects dropped out of the program at the beginning due to personal decisions. Figure 2 shows the CONSORT flow diagram. Baseline characteristics of the participants are shown in Table 1.

### 3.2. Body Composition

Analysis of body composition showed that there was no significant decrease, or notable effect size, in FM for either group (*p* = 0.212, ES = −0.2 and *p* = 0.389, ES −0.1 for TT and UT, respectively); similarly, no significant change in FFM was present for TT (*p* = 0.679, ES = 0.0) or UT (*p* = 0.145, ES = 0.1). However, there was an increase in FFM-FFAT in both groups (*p* = 0.012, ES = 0.5 and *p* = 0.001, ES = 0.7 for TT and UT, respectively). In BMD, no significant changes were found in TT (*p* = 0.745, ES = 0.0) or UT (*p* = 0.844, ES = 0.0). Comparison analysis between groups showed that there were no differences in these body composition variables (Table 2 and Figure 3).

### 3.3. Functional Capacity

The results on strength-related variables showed that both groups had significant increases and a large effect size in row (TT, *p* ≤ 0.01, ES = 1.3, UT, *p* ≤ 0.01, ES = 1.1) and bench press (TT, *p* < 0.01, ES = 1.3, UT, *p* < 0.01, ES = 1.5). Regarding lower extremity strength, significant increases in LE were found in both groups, but with a large effect size in TT (*p* < 0.01, ES = 1.2) and medium in UT (*p* ≤ 0.01, ES = 0.5); on the other hand, for strength assessed by HLP-M, significant increases and a large effect size were present in both groups, especially for UT (TT, *p* = 0.001, ES = 1.0, UT, *p* = 0.007, ES = 1.5). Squat performance presented significant increases and medium effect sizes in TT and UT (*p* = 0.008, ES = 0.6 and *p* < 0.01, ES = 0.4, respectively). HGS resulted in significant increases in both groups, but the effect size was slightly higher in TT than UT (*p =* 0.006, ES = 0.4 and 0.014, ES = 0.1, for TT and UT, respectively). Strength in elbow flexors, examined by ACT, resulted in a significant increase and a large effect size in both groups (*p* = 0.008, ES = 1.3 and *p* < 0.01, ES = 1.6 for TT and UT, respectively). CST showed significant increases and a large effect size for both groups (*p* = 0.008, ES = 1.3, UT, *p* < 0.01, ES = 1.8). Regarding range of motion measured by CSRT, no changes were found for either group (TT, *p* = 0.674, ES = −0.3, UT, *p* = 0.525, ES = 0.1, respectively), and no changes in BST were found in any of the groups (TT, *p* = 0.171, ES = 0.25, UT = 0.017, ES = 0.20). Cardiorespiratory fitness assessed through the 6 MWT revealed significant increases with a large effect size in TT (*p* < 0.01; ES = 2.7) and UT (*p* = 0.006, ES = 2.2). On the other hand, agility/dynamic balance assessed by the 8-FUG showed no changes in UT (*p* = 0.559, ES = 0.2) and negative changes in TT (*p* = 0.003, ES = −0.8). Finally, the comparison between groups showed differences in HGS (*p* = 0.011), MR (*p* = 0.041), LE (*p* = 0.001), and 6 MWT (*p* = 0.010) in favor of TT, and a difference was found in 8-FUG (*p* = 0.004) to the detriment of TT (Table 2, Figure 4 and Figure 5).

## 4. Discussion

This study aimed to evaluate the impact of two 8-week strength protocols with traditional or undulating periodization on body composition, strength-related variables, and physical fitness parameters in male and female HOA. The main findings of our study are that an 8-week UT or TT protocol seem valid for improving FFM and increasing strength and functional capacity in women and men over 60 years old. Similar to our results, some studies have found significant increases in FFM after a RT intervention using DXA for monitoring body composition. For example, a 12-week RT program augmented FFM in men aged 54–71 years [30]. In addition, while nine weeks have failed to improve FFM [31], three days of RT per week for 24 weeks have been demonstrated to significantly increase FFM in participants aged 65–75 years [32]. However, other research studies have found contrasting results. For example, Martel et al. found no significant increase in FFM in HOA aged 65–73 years after applying a protocol of various upper- and lower-limb exercises three days per week for 24 months [33]. Likewise, Hurlbut et al. reported no changes after a similar RT protocol for 24 weeks in older participants between 65–75 years [34]. Finally, significant improvement in FFM has been shown in men but not in women of 65–75 years after 24 weeks [35]. We highlight the fact that most of the investigations did not correct FFM for FFAT, which might have influenced interpretation and conclusions. In fact, several studies have highlighted the importance of correcting FFM for this fat-free component of the adipose tissue to have more accurate outcomes regarding changes in body composition in older adults [36,37]. 

Technical issues of the DXA measurement may influence the results when evaluating musculoskeletal mass since the cross-sectional area was not directly measured. Previous research has shown positive effects on cross-sectional area assessed by computed tomography after strength training protocols for 10 months in HOA between 60–80 years [38]. Similarly, evaluations performed with nuclear magnetic resonance have also shown significant differences in muscle volume after three training days per week of quadriceps extension for 9 weeks in participants aged 65–75 years [39]. Moreover, increases in the cross-sectional area have been found after 12 weeks of training in older adults between 50 and 70 years [40]. Finally, the use of muscle biopsy has revealed significant increases in the vastus lateralis of the quadriceps after 24 [41] or 26 weeks [42] of exercise training. Pyka et al. also found an augmentation in the cross-sectional area after 15 and 30 weeks of an exercise protocol consisting of 3 sets of 8 repetitions at 75% RM three days per week [43]. Notwithstanding, inter-individual variability and sex-based differences should be taken into account. Using the muscle biopsy method, Lexell et al. did not show significant differences in the biceps brachii cross-sectional area, and even reductions by 14% were found in the vastus lateralis, after a strength training program in men aged 70–77 years [44]. Older women might be less susceptible to increases in FFM as has been shown previously [30,34]. Since our sample included four female participants in each group (4/9, 44%), further research is needed to explore sex-dependent differences.

We did not find changes in FM, which may have been due to the initial nutritional recommendations of energy surplus. This coincides with the studies conducted by Hurlbut et al. [34] and Ryan et al. [35], where no changes in FM were detected in any of the evaluated groups after strength training. It is worth noting that Joseph et al. reported reductions in FM [30]. Such disparate differences in body composition warrant more research to establish definitive outcomes after a give RT protocol. Likewise, although we did not find changes in bone mass after the exercise intervention, future studies may evaluate the long-term effects of a strength training program (>6 months) considering the positive changes on bone mineral density that have been reported previously [45]. 

In regards to strength-related variables, our findings suggest that eight weeks of RT have positive effects in both HGS, albeit a higher clinical significance was detected for the TT group (ES = 0.4 and ES = 0.1 for TT and UT, respectively). It is important to point out that we have not differentiated between the force generated by the dominant and non-dominant hand, since previous research has shown higher force increases in the dominant hand [46]. We also found significant increases in upper- and lower-limb strength values after both TT and UT protocols; for example, the squat 1 RM increased significantly (*p* = 0.008, ES = 0.6 and *p* < 0.01, ES = 0.4, respectively) with no statistical differences between groups. Our findings are in agreement with previous findings that have demonstrated: (i) the positive impact of strength training on the squat 1 RM in >60-year-old participants [47], (ii) no differences between the TT and UT protocols on muscle power and strength after 12 weeks in HOA [22], and (iii) similar improvements in lower-limb strength after 16 weeks of TT and UT programs in older women [48]. Finally, the outcomes of the SFT did not show improvements in agility, balance, or flexibility, which is in agreement with previous findings [48]; however, the TT and UT protocols positively impacted on cardiorespiratory fitness and revealed high clinical significance (ES = 2.7 and ES = 2.2, respectively). 

Our study has several limitations that need to be mentioned. Firstly, the sample size does not allow for generalizing the findings. Secondly, although we gave nutritional recommendations to the participants at the beginning of the study, no rigorous nutritional control was performed. Adherence was not assessed either. These facts might have influenced body composition and, thereby, more controlled studies are needed to evaluate the concurrent effects of UT, high-protein diet, and energy surplus as a potential tool to improve morphological and strength outcomes. It needs to be noted that we corrected FFM for FFAT in order to have more accurate interpretations on body composition changes. Furthermore, even though an unsupervised nutritional approach was implemented, our training protocols were able to generate positive changes on FFM and physical function.

## 5. Conclusions

Our results suggest that an eight-week period of strength training with or without an undulating approach might be a viable option to improve FFM and increase strength levels in HOA. In addition, cardiorespiratory fitness and functional capacity were significantly enhanced by the strength training protocol regardless of whether UT or TT were implemented. Future studies should analyze inter-individual variability and a sex-dependent differences section is mandatory.

## 6. Practical Application

The implementation of UT programs may be a new option for the improvement of strength levels in older adults, as well as other health markers. New combinations of configurations should be studied to seek the greatest effectiveness of these training programs.

## Figures and Tables

**Figure 1 ijerph-19-04522-f001:**
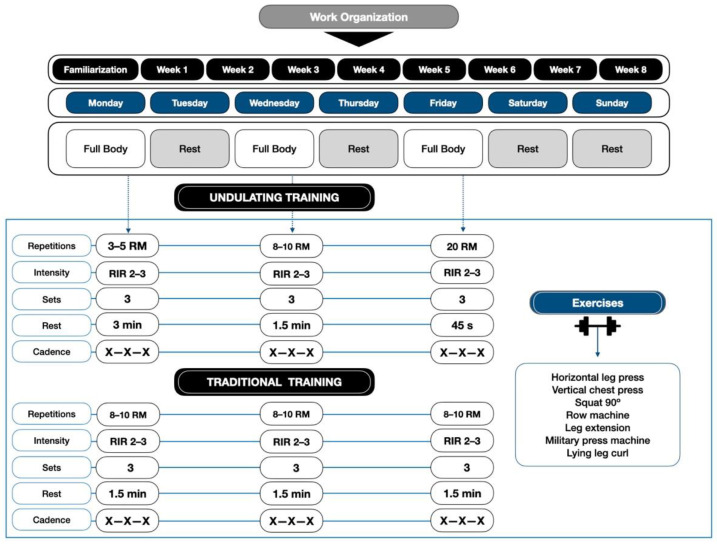
Resistance Training Organization. X-X-X = maximal intended velocity in the concentric, eccentric, and isometric phases; RIR = repetitions in reserve.

**Figure 2 ijerph-19-04522-f002:**
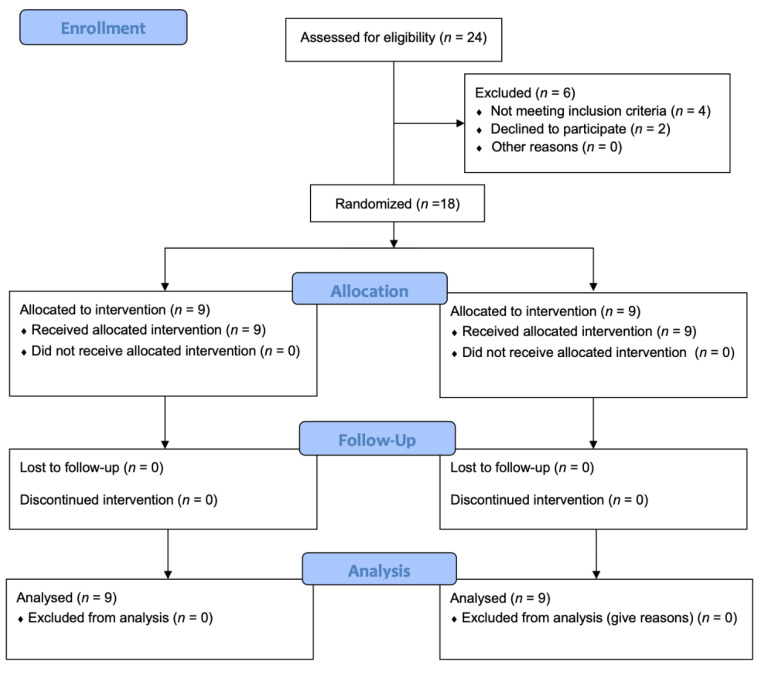
Consolidated Standards of Reporting Trials (CONSORT) flow diagram.

**Figure 3 ijerph-19-04522-f003:**
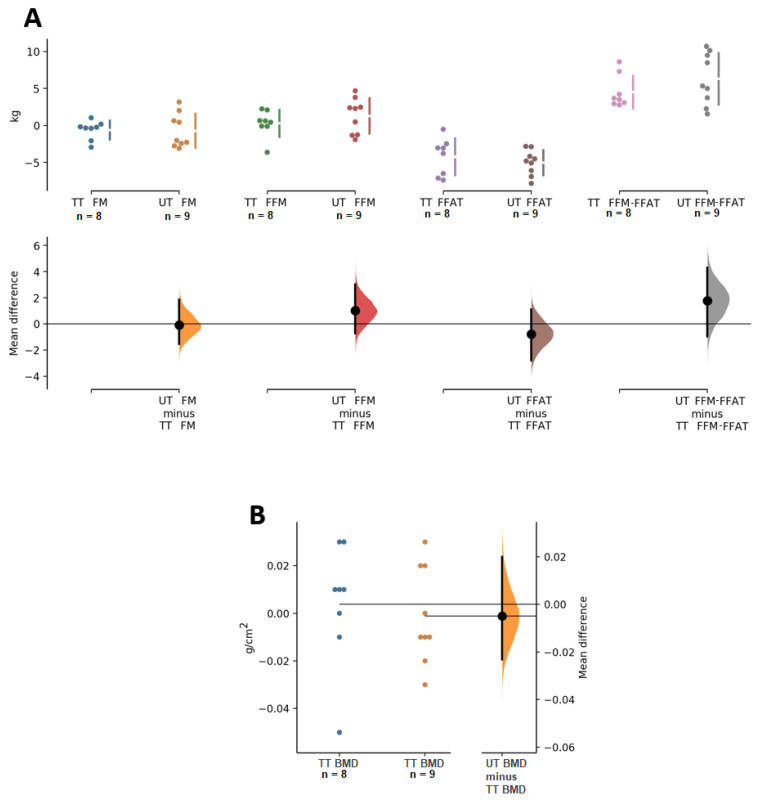
Paired Cumming estimation plots of body composition variables. (**A**) FM, fat mass; FFM, fat free mass; FFAT, Fat-free adipose tissue, and FFM-FFAT, fat free mass corrected for fat-free adipose tissue. (**B**) BMD, bone mineral density. TT, traditional training; UT, undulating training. Both groups are plotted on the left axes; the mean difference is plotted on a floating axis on the right as a bootstrap sampling distribution. The mean difference is depicted as a dot; the 95% confidence interval is indicated by the ends of the vertical error bar [28].

**Figure 4 ijerph-19-04522-f004:**
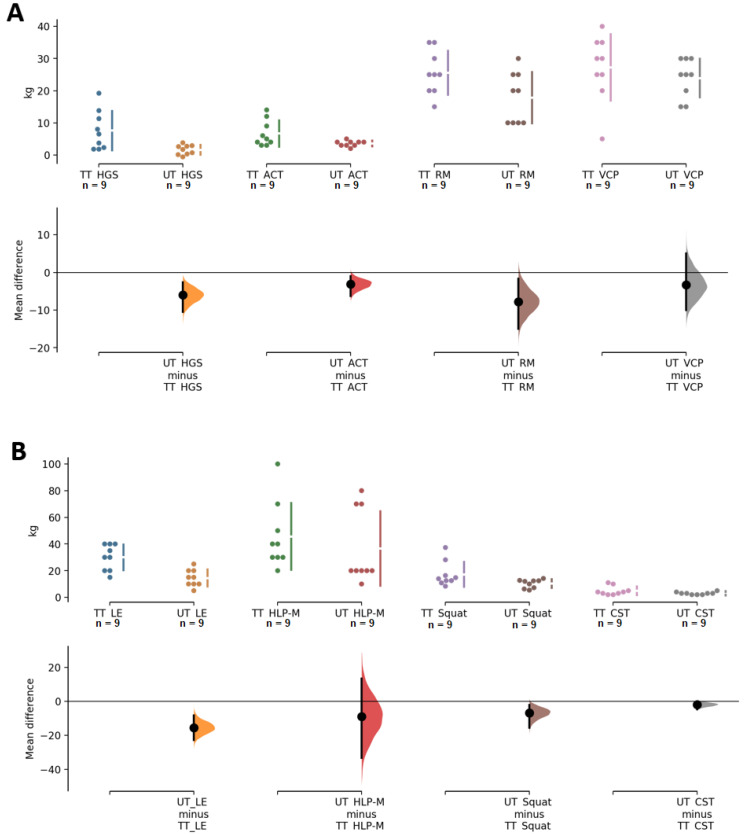
Paired Cumming estimation plots of upper- and lower-limb strength. (**A**) HGS, handgrip strength; ACT, 1 RM-Arm Curl test; RM, 1RM-Row machine, and VCP, 1 RM-vertical chest press (**B**) LE, 1 RM-Leg extension; HLP-M, 1 RM-Horizontal leg press-Monopodal; Squat; CST, and Chair stand test. TT, traditional training; UT, undulating training. Both groups are plotted on the left axes; the mean difference is plotted on a floating axis on the right as a bootstrap sampling distribution. The mean difference is depicted as a dot; the 95% confidence interval is indicated by the ends of the vertical error bar [28].

**Figure 5 ijerph-19-04522-f005:**
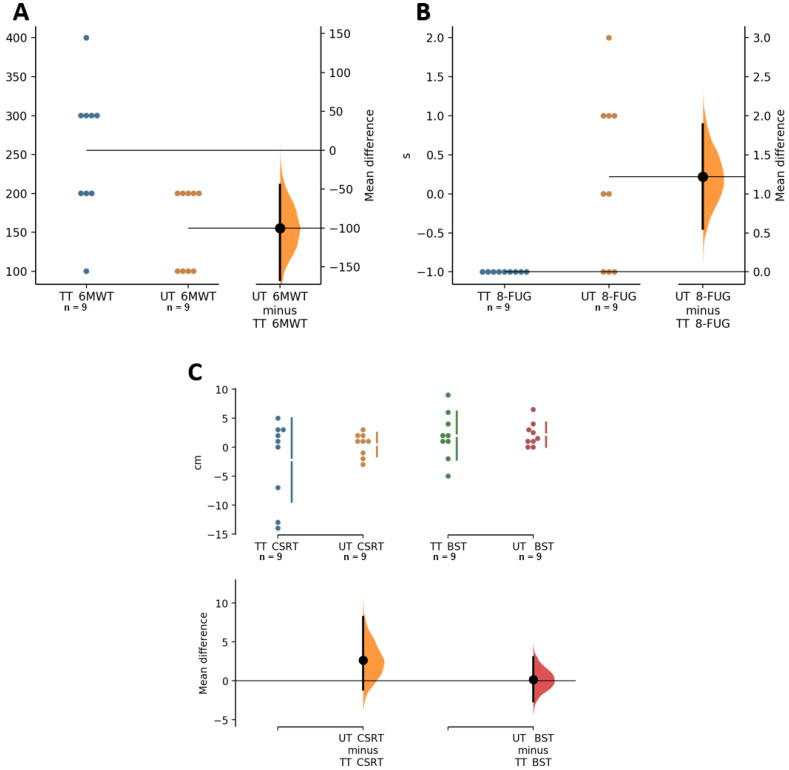
Paired Cumming estimation plots of functional capacity. (**A**) 6 MWT, 6-min six walk test. (**B**) 8-FUG, 8 Foot Up-and-Go Test. (**C**) CSRT, chair sit and reach test, and BST, back scratch test. TT, traditional training; UT, undulating training. The groups are plotted on the left axes; the mean difference is plotted on a floating axis on the right as a bootstrap sampling distribution. The mean difference is depicted as a dot; the 95% confidence interval is indicated by the ends of the vertical error bar [28].

**Table 1 ijerph-19-04522-t001:** Characteristics of the participants.

Variables	TT (*n* = 9)	UT (*n* = 9)	*p*-Level
Age (years)	64.3 ± 2.1	63.7 ± 2.1	0.514
Stature (cm)	165.4 ± 7.9	164.9 ± 7.5	0.890
BM (kg)	70.2 ± 9.5	72.4 ± 12.1	0.681
BMI (kg·m^−2^)	25.7 ± 1.8	26.6 ± 3.9	0.570
FM (kg)	21.6 ± 3.5	22.3 ± 5.1	0.724
FFM (kg)	48.6 ± 9.8	50.1 ± 9.4	0.762
FFAT (kg)	7.9 ± 2.3	8.8 ± 1.7	0.359
FFM-FFAT (kg)	40.7 ± 8.6	41.2 ± 7.8	0.893
BMD (g·cm^−2^)	1.1 ± 0.1	1.1 ± 0.1	0.576
HGS (kg)	66.7 ± 17.6	66.3 ± 18.4	0.961
ACT (kg)	21.3 ± 3.9	17.2 ± 2.0	0.017
RM (kg)	56.7 ± 16.6	48.3 ± 12.2	0.243
VCP (kg)	38.9 ± 20.1	32.8 ± 15.0	0.476
LE (kg)	62.2 ± 19.4	62.2 ± 23.2	1.000
HLP-M (kg)	71.1 ± 36.6	83.3 ± 27.8	0.437
Squat (kg)	60.9 ± 24.4	60.1 ± 24.7	0.947
CST (kg)	13.3 ± 2.3	13.6 ± 1.4	0.808
CSRT (kg)	−2.6 ± 9.4	−3.6 ± 6.3	0.795
BST (cm)	10.8 ± 9.1	11.2 ± 9.1	0.919
6 MWT (m)	533.3 ± 50.0	477.8 ± 44.1	0.024
8-FUG	6.4 ± 1.2	6.2 ± 1.1	0.692

Data are expressed as mean ± standard deviation. TT, traditional training; UT, undulating training; BM, body mass; BMI, body mass index; FM, fat mass; FFM, fat free mass; FFAT, fat-free adipose tissue; FFM-FFAT, fat free mass corrected for fat-free adipose tissue; BMD, bone mineral density; HGS, handgrip strength; ACT, 1 RM-Arm Curl test; RM, 1RM-Row machine; VCP, 1 RM-vertical chest press; LE, 1 RM-leg extension; HLP-M, 1 RM-horizontal leg press-monopodal; CST, chair stand test; CSRT, chair sit and reach test; BST, back scratch test; 6 MWT, 6-min six walk test; 8-FUG, 8 foot Up-and-Go Test.

**Table 2 ijerph-19-04522-t002:** Pre- and post-intervention data on the main study variables.

	TT			TU			Between-Group Difference	
	Δ ± SD (95% CI)	*p*	ES	Δ ± SD (95% CI)	*p*	ES	TT − TU	*p*
FM (kg)	−0.6 ± 1.3 (−1.7–0.4)	0.212	−0.2	−0.7 ± 2.3 (−2.5–1.1)	0.389	−0.1	0.1 (−1.9–2.0)	0.928
FFM (kg)	0.3 ± 1.8 (−1.2–1.8)	0.679	0.0	1.3 ± 2.4 (−0.5–3.1)	0.145	0.1	−1.0 (−3.2–1.2)	0.346
FFAT (kg)	−4.2 ± 2.5 (−6.3–−2.2)	0.002	−2.5	−5.0 ± 1.7 (−6.3–−3.7)	<0.01	−3.5	0.8 (−1.4–3.0)	0.459
FFM-FFAT (kg)	4.5 ± 2.2 (2.7–6.4)	0.012	0.5	6.3 ± 3.5 (3.6–9.0)	0.001	0.7	−1.8 (−4.9–1.3)	0.232
BMD (g·cm^2^)	0.00 ± 0.03 (−0.02–0.02)	0.745	0.0	0.00 ± 0.02 (−0.02–0.01)	0.844	0.0	0.00 (−0.02–0.03)	0.697
HGS (kg)	7.6 ± 6.1 (2.9–12.3)	0.006	0.4	1.6 ± 1.5 (0.4–2.7)	0.014	0.1	6.0 (1.6–10.5)	0.011
ACT (kg)	6.7 ± 4.1 (3.5–9.8)	0.001	1.1	3.6 ± 0.9 (2.9–4.2)	<0.01	1.6	3.1 (0.2–6.0)	0.056
RM (kg)	25.6 ± 6.8 (20.3–30.8)	<0.01	1.3	17.8 ± 7.9 (11.7–23.9)	<0.01	1.1	7.8 (0.4–15.2)	0.041
VCP (kg)	27.2 ± 10.3 (19.3–35.2)	<0.01	1.3	23.9 ± 6.0 (19.3–28.5)	<0.01	1.5	3.3 (−5.1–11.8)	0.415
LE (kg)	30.0 ± 9.7 (22.6–37.4)	<0.01	1.2	14.4 ± 6.3 (9.6–19.3)	<0.01	0.5	15.6 (7.4–23.7)	0.001
HLP-M (kg)	45.6 ± 25.1 (26.3–64.8)	0.001	1.0	36.7 ± 27.8 (15.3–58.1)	0.007	1.5	8.9 (−17.6–35.4)	0.487
Squat (kg)	17.2 ± 9.4 (10.0–24.4)	0.008	0.6	10.3 ± 3.2 (7.8–12.7)	<0.01	0.4	6.9 (−0.1–13.9)	0.053
CST (kg)	4.9 ± 3.3 (2.3–7.5)	0.008	1.3	3.0 ± 1.0 (2.2–3.8)	<0.01	1.8	1.9 (−0.6–4.3)	0.202
CSRT (kg)	−2.2 ± 7.2 (−7.8–3.3)	0.674	−0.3	0.4 ± 2.0 (−1.1–2.0)	0.525	0.1	−2.7 (−8.0–2.6)	0.302
BST (cm)	2.0 ± 4.1 (−1.2–5.2)	0.171	0.25	2.0 ± 2.0 (0.5–3.5)	0.017	0.20	0.0 (−3.2–3.2)	1.00
6 MWT (m)	255.6 ± 88.2 (187.8–323.3)	<0.01	2.7	155.6 ± 52.7 (115.0–196.1)	0.006	2.2	100.0 (27.4–172.6)	0.010
8-FUG	−1.0 ± 0.0 (−1.0–−1.0)	0.003	−0.8	0.2 ± 1.1 (−0.6–1.1)	0.559	0.2	−1.2 (−2.0–−0.4)	0.004

Data are expressed as mean change (Δ) ± standard deviation (SD) and the corresponding 95% confidence interval (95% CI), and effect size (ES). TT, traditional training; UT, undulating training; BM, body mass; BMI, body mass index; FM, fat mass; FFM, fat free mass; FFAT, fat-free adipose tissue; FFM-FFAT, fat free mass corrected for fat-free adipose tissue; BMD, bone mineral density; HGS, handgrip strength; ACT, 1 RM-Arm Curl test; RM, 1RM-Row machine; VCP, 1 RM-vertical chest press; LE, 1 RM-leg extension; HLP-M, 1 RM-horizontal leg press-monopodal; CST, chair stand test; CSRT, chair sit and reach test; BST, back scratch test; 6 MWT, 6-min six walk test; 8-FUG, 8 foot Up-and-Go Test.

## Data Availability

Data are available for educational purposes with approval from the authors as long as provision of the data does not violate any university, sponsor, or institutional review board policies.

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
