# Peer review of "Traditional and Undulating Periodization on Body Composition, Strength Levels and Physical Fitness in Older Adults"

_ijerph, 2022, doi:10.3390/ijerph19084522_

Round 1
Reviewer 1 Report
First of all, I want to thank the opportunity to review this work, as well as congratulate the authors for the quality and interesting scope of the research.
The purpose of this work was to evaluate body composition, strength levels, and physical fitness in response to traditional or undulating training in older adult, which is an interesting topic of current development.
The document is well written, structured and offers a robust analysis to take into account the results obtained. Some observation and recommendation is made in order to facilitate the information to the readers.
1. In relation to the more formal structure, it is recommended to check some typographical errors throughout the text, especially regarding the end of the line (for example, line 37, 54, 70, etc.)
2. In the abstract it is recommended to include some reference on the selected statistical analysis, and type study (double-arm and repeated-measures randomized clinical trial) to facilitate the understanding in this section of the analysis carried out.
3. It is assumed that, initially, 24 participants were selected, but only 18 carried out the study (it is recommended to clarify this in sections 2.1 and 2.2, to adapt the data to table 1, where it is clearly explained); In addition, in the text (for example in the abstract)reference is already made to individuals over 60 years of age, but the participants have a range between 58 (is it assumed that no one under 60 years of age was selected?) and 65 years of age ("A total of eighteen older adults, 10 male and 8 female, 64±2.1 years"). Besides, the mean age in table 1 reports 64.3 +/- 2.1 and 63.7 +/- 2.1. It is recommended to adjust the information (and data) in the parts of the text where there may be a possible difference.
4. The most important aspect that I wish to highlight, and necessary to include in section 2.4, is in reference to how the procedure was carried out to establish the different RMs used. In my opinion, and taking into account the selected population, this is a very important methodological part to know how the knowledge of the loads that were later used to perform the intervention was carried out. In summary, the intervention protocols are explained, but the load adjustment protocols with respect to the different selected RMs are not explained.
This is my major problem and it is necessary to include it in order to understand and/or reproduce the design in other studies.
5. Additionally, it is necessary to clarify what type of movement was selected, that is, eccentric and concentric, or only concentric ("Both protocols were performed with maximal execution speed, mainly in the concentric phase"). The information appears in the legend of figure 1, but a detailed explanation of the methodology must be explained in the corresponding section.
6. Line 172: 72.5±11.4; kg Put "comma" instead of "semicolon" (72.5±11.4, kg)
7. The data shown in lines 171-172 (specifically 165.12±7.5 cm, 171 72.5±11.4; kg, 26.5±3.2 k gm-2) do not coincide with those shown in Table 1, if the means are checked; Please adjust the data for accuracy.
8. The columns of table 2 must be adjusted for a better visualization of the data.
Author Response
Many thanks for your comments.

Reviewer 2 Report
Thank you for taking the time to complete the research in the area of older adults, traditional training and undulating training. This was reviewed with interest. I have a few questions regarding the manuscript along with some comments to hopefully help create a more succinct version for the readers of the journal
- In the introduction the authors are very clear in what TT training is, but there is no clear viewpoint of what is meant by UT training. Please update this part of the manuscript.
- In the methods section it may be helpful to the reader to understand the fitness tests that were used for the participants before the exercise programming information. I would suggest moving 3.2, 3.3, and 3.4 before 2.3 and 2.4. 3.1 can go following 2.4. With 3.1 how do you know that the dietary recommendations were followed if participants did not complete dietary diaries.
- The statistics seem appropriate for the study.
- In the results section you note that there were 10 males and 8 females, but there is no indication of how many males and females were in each group. Please tend to this.
- It would be helpful if the tables were split into two - DEXA/Anthropometry as table 1, and table 2 senior fitness test.
- It would be helpful if the abbreviations followed the list in the table instead of alphabetical.
- Please discuss why there were differences between the two groups at the start in terms of ACT and 6MWT, was this simply that there were more males in TT than UT?
- This difference was further widened between TT and. UT following the exercise programme, but this was not discussed.
- At the end of the programme, even though HGS, ACT, LE, SQT, 6MWT and TUG test were all improved, they seem to be more improved with TT than UT. Why do you think this was?
- There was no ethical information regarding the study. This should be provided in the methods section.
Author Response
Thank you very much for your comments.

Reviewer 3 Report
Dear editor and authors, I thank you for the invitation to evaluate this manuscript. My question concerns the practical application of the findings of this research. I request the authors to add this point in the manuscript, also mentioning the potential risks and benefits of this type of training, which in the elderly can be potentiated due to the decrease of some physiological, morphological, and even functional functions.
Author Response
Thank you very much for your comments.
Round 2
Reviewer 1 Report
I want to thank the authors for their attention to the suggested changes, as well as congratulate them for the work done on its current version.
All proposals for improvement have been addressed and, in my opinion, except for better interpretation, their quality allows publication.
I only want to indicate that, in the final document, the configuration of table 2 be taken into account, since it seems to be misconfigured when passing to the platform format.
Best regards.